Evidence from studies in rodents and in isolated adipocytes that agonists of the chemerin receptor CMKLR1 may be beneficial in the treatment of type 2 diabetes

Wargent Edward T. 1
Zaibi Mohamed S. 1
O’Dowd Jacqueline F. 1
Cawthorne Michael A. 1
Wang Steven J. 2 *
Arch Jonathan R.S. 1 jon.arch@buckingham.ac.uk
Stocker Claire J. 1
1 Clore Laboratory, Buckingham Institute for Translational Medicine, University of Buckingham , Buckingham , UK
2 AstraZeneca R & D, Alderley Park , Macclesfield , UK
Liu Jie
* Current affiliation: Sheffield Healthcare Gateway, The University of Sheffield, Sheffield, UK

Electronic publication date: 2015 Feb 5
Publication date: 2015
Volume: 3
Electronic Location ID: e753
Received 2014 Dec 6; Accepted 2015 Jan 18
Copyright: © 2015 Wargent et al.
Copyright year: 2015
Copyright holder: Wargent et al.
License: This is an open access article distributed under the terms of the Creative Commons Attribution License, which permits unrestricted use, distribution, reproduction and adaptation in any medium and for any purpose provided that it is properly attributed. For attribution, the original author(s), title, publication source (PeerJ) and either DOI or URL of the article must be cited.
License URL: https://creativecommons.org/licenses/by/4.0/

Keywords: Type 2 diabetes, Glucose uptake, Chemerin receptor, Adipocyte, Obesity, Chemerin, Rosiglitazone

Funding: AstraZeneca 20080509LL/SEML-7DSHX3 Clore Laboratory AstraZeneca funded most of the work described in this manuscript (grant 20080509LL/SEML-7DSHX3). Some of the glucose uptake work was funded from the resources of the Clore Laboratory after the formal collaboration with AstraZeneca ended. The original idea to work in this area was Steven Wang’s. He had moved from the Clore Laboratory to work for AstraZeneca, who had chemerin receptor knockout mice available to them. The details of the experiments were designed by University of Buckingham authors but discussed with Steven Wang to solicit his input. The funders had no role in study design, data collection and analysis, decision to publish, or preparation of the manuscript.

==============================
The literature is unclear on whether the adipokine chemerin has pro- or anti-inflammatory properties or plays any role in the aetiology of type 2 diabetes or obesity. To address these questions, and in particular the potential of agonists or antagonists of the chemerin receptor CMKLR1 in the treatment of type 2 diabetes and obesity, we studied the metabolic phenotypes of both male and female, CMKLR1 knockout and heterozygote mice. We also investigated changes in plasma chemerin levels and chemerin gene mRNA content in adipose tissue in models of obesity and diabetes, and in response to fasting or administration of the insulin sensitizing drug rosiglitazone, which also has anti-inflammatory properties. The effects of murine chemerin and specific C-terminal peptides on glucose uptake in wild-type and CMKLR1 knockout adipocytes were investigated as a possible mechanism by which chemerin affects the blood glucose concentration. Both male and female CMKLR1 knockout and heterozygote mice displayed a mild tendency to obesity and impaired glucose homeostasis, but only when they were fed on a high-fat died, rather than a standard low-fat diet. Obesity and impaired glucose homeostasis did not occur concurrently, suggesting that obesity was not the sole cause of impaired glucose homeostasis. Picomolar concentrations of chemerin and its C15- and C19-terminal peptides stimulated glucose uptake in the presence of insulin by rat and mouse wild-type epididymal adipocytes, but not by murine CMKLR1 knockout adipocytes. The insulin concentration-response curve was shifted to the left in the presence of 40 pM chemerin or its C-15 terminal peptide. The plasma chemerin level was raised in diet-induced obesity and ob/ob but not db/db mice, and was reduced by fasting and, in ob/ob mice, by treatment with rosiglitazone. These findings suggest that an agonist of CMKLR1 is more likely than an antagonist to be of value in the treatment of type 2 diabetes and to have associated anti-obesity and anti-inflammatory activities. One mechanism by which an agonist of CMKLR1 might improve glucose homeostasis is by increasing insulin-stimulated glucose uptake by adipocytes.

Introduction

Chemerin is translated as a 163 amino acid pre-proprotein and secreted as a 143 amino acid proprotein. Further C-terminal cleavage by extracellular proteases results in peptides that have chemoattractant properties. Some experimental evidence shows that these peptides have pro-inflammatory properties, but there is also evidence showing that some have anti-inflammatory properties (Ernst & Sinal, 2010). The C-15 peptide is reported to have potent anti-inflammatory activity, and the C-19 peptide is reported to be inflammatory but to lack any chemotactic activity for macrophages (Cash et al., 2008). We (and collaborators) have shown that chemerin also stimulates angiogenesis (Bozaoglu et al., 2010).

The possibility that chemerin plays a role in metabolic disease, in particular with obesity and type 2 diabetes, arose when it was discovered that adipose tissue is one of the main tissues from which chemerin is secreted (i.e., it is an adipokine), and that circulating levels in humans correlate with body mass index, plasma triglycerides and blood pressure (Ernst & Sinal, 2010). Insulin sensitising thiazolidinedione drugs that activate peroxisome proliferator-activated receptor (PPAR) γ alter the expression of chemerin and circulating chemerin levels, but contrasting effects have been reported (Ernst & Sinal, 2010; Roman, Parlee & Sinal, 2012). For example, amelioration of insulin resistance and hyperglycaemia by the PPARγ agonist pioglitazone or by metformin in patients with type 2 diabetes is accompanied by reduced plasma chemerin (Esteghamati et al., 2014), whereas treatment of mice with the PPARγ agonist rosiglitazone increased chemerin mRNA levels in adipose tissue and circulating chemerin levels (Muruganandan et al., 2011).

Since chemerin levels are raised in obesity, a pro-inflammatory effect might be one mechanism by which obesity causes insulin resistance. Mechanisms that are independent of inflammation might also play a role, because in some studies chemerin has inhibited glucose uptake too rapidly to suggest a role for inflammation, or inhibition of glucose uptake has been observed in isolated cell systems. In addition, chemerin promotes adipogenesis, which might exacerbate obesity (Ernst & Sinal, 2010; Roman, Parlee & Sinal, 2012). However, it is by no means certain that compounds that block the effects of chemerin would be of value in the treatment of metabolic disease. For example, inhibitory as well as stimulatory effects of chemerin on glucose uptake in 3T3-L1 adipocytes have been described (Takahashi et al., 2008; Kralisch et al., 2009). Thus, elevated plasma levels of chemerin might play a causal role in obesity and its metabolic complications, but it is also possible that they play a counter-regulatory role. In particular, chemerin might mitigate insulin resistance by permitting adipose tissue expansion through its angiogenic effect and by promoting adipogenesis (Bozaoglu et al., 2010).

The first chemerin receptor to be identified is known as CMKLR1, or ChemR23. It is highly expressed in adipose tissue, in various cell types involved in innate and adaptive immunity, and in endothelial cells (Ernst & Sinal, 2010; Kaur et al., 2010). Adipose tissue is therefore both a major source of chemerin and a site of its action. Studies conducted in mice that lack this receptor have revealed a complex and inconsistent metabolic phenotype. Both beneficial and adverse age-, sex- and diet-related effects on body composition and glucose homeostasis have been described, as well as an absence of these effects (Ernst et al., 2012; Rouger et al., 2013; Gruben et al., 2014).

Here we have followed four main lines of inquiry. First, we show that in our hands both male and female mice that lack CMKLR1 and are fed on a high-fat diet have a mild tendency to increased body fat and impaired glucose homeostasis, though these do not always occur at the same age. We are the first to describe similar results in mice that are heterozygote for CMKLR1. Second, we show for the first time that murine chemerin and its C-15 and C-19 terminal peptides stimulate glucose uptake in primary adipocytes, and provide evidence that this effect is mediated through CMKLR1. Third, we investigate the effects of fasting and high fat diet-induced obesity on plasma chemerin levels and chemerin gene (RRARES2) mRNA content in adipose tissue, comparing for the first time the effects of diet-induced obesity on plasma chemerin levels in C56Bl/6 and FVB mice, the latter being relatively resistant to obesity. Fourthly, we report that, as in wild-type mice (Muruganandan et al., 2011), rosiglitazone increases the expression of the chemerin gene RARRES2 in inguinal adipose tissue and raises the plasma chemerin level in leptin deficient ob/ob mice. Our findings are consistent with chemerin, through its receptor CMKLR1, having beneficial effects in type 2 diabetes, which is contrary to the opinion of some other authors. Thus, an agonist of CMKLR1 might be effective in the treatment of type 2 diabetes.

Materials and Methods

Materials

Murine chemerin was purchased from R&D Systems Europe Ltd (Abingdon, Oxfordshire, UK). Its C-15 (amino acids 140–154), C-19 (amino acids 138–156) and stable analogue of C-9 (amino acids 148–156) (Shimamura et al., 2009; Cash et al., 2008) terminal peptides were custom-synthesized by Cambridge Research Biochemicals (Billingham, Cleveland, UK).Their amino acid sequences were:

C-15: AGEDPHGYFLPGQFA

C-19: AQAGEDPHGYFLPGQFAFS

C-9: yFLPsQFaTicS (This is a modification of the C-9 sequence FLPGQFAFS, with D-Tyr147, D-Ser151, D-Ala154, Tic155. Although we refer to it as C-9 analogue because it was first described as an analogue of C-9, the introduction of D-Tyr147 actually makes it a C-10 peptide.)

Other reagents were obtained from Sigma-Aldrich, Poole, UK, unless otherwise stated.

Animals

Mice and rats were from Charles Rivers (Maidstone, Kent, UK), except for the embryos used to generate the CMKLR1 knockout and heterozygote mice, which were generated by Deltagen (see below). Mice were received at five weeks of age and rats at six weeks of age (150 g body weight). They were fed on standard laboratory chow (Beekay Feed; B&K Universal Ltd., Hull, UK) until used, except that for the studies on diet-induced obesity male C57Bl/6 and FVB mice were fed from the age of 6 weeks on a high-fat (63% by energy; Open Source D12492, Research Diets, New Brunswick, NJ, USA) diet for 6 months. Housing and procedures were conducted in accordance with the UK Government Animal (Scientific procedures) Act 1986 and approved by the University of Buckingham Ethical review Board. Animals were killed 3–4 h after the onset of the light cycle, by a UK Government Animal Scientific Act 1986 schedule 1 method.

Whole body CMKLR1 knockout mice were originally generated by Deltagen (San Mateo, CA, USA) and supplied as embryos to AstraZeneca, Alderley Park, where they were backcrossed onto the C57Bl/6 background for three generations. Heterozygote mice were supplied to the University of Buckingham and bred for a minimum of five further generations to generate the animals for the phenotyping experiments. Genotyping of experimental animals was performed in accordance with the recommendations of Deltagen.

Adipocytes were prepared from mesenteric fat pads of female wild-type C57Bl/6 mice, epididymal fat pads of male Sprague-Dawley rats (350–400 g), and epididymal fat pads of male wild-type and CMKLR1knockout C57Bl/6 mice by a method that we have described previously (Zaibi et al., 2010). All animals were 12–14 weeks old.

The effect of fasting for 16 h overnight was studied in 8-week-old male wild-type C57BL/6J mice. The effect of treatment with rosiglitazone (3 mg/kg body weight, p.o., once daily; Sequoia Research Products Ltd, Japan) for three weeks and of fasting for 16 h overnight was studied in female ob/ob mice. Male db/db mice and wild-type (misty C57Bl/Ks) mice were 11–12 weeks old when studied.

Phenotyping experiments

Food intake and body weight were measured weekly. Non-fasting blood glucose was measured at 9 weeks, and 3 and 6 months of age. Blood glucose following a 5 h fast was measured at 8 weeks, 11 weeks and 6 months of age. Insulin tolerance was measured at 6 months of age. Body composition and glucose tolerance were conducted at 7–8 weeks and 3 and six months of age.

Body composition was measured under light gaseous anaesthetic (Isofluorane; Isoba, Shering-Plough Animal Health, UK) using dual-energy X-ray absorptiometry (Lunar PIXImus 2 mouse densitometer and version 1.46 software; GE Medical Bedford, UK).

Oral glucose tolerance was measured after mice had been fasted for five hours before being dosed with glucose (3 g/kg body weight, p.o.). Blood samples (10 µl) were taken from the tip of the tail after applying a local anesthetic (Lignocaine™; Centaur Services, UK), 30 min and immediately before, and 30, 60, 90, 120 and 180 min after dosing the glucose load. Whole blood was mixed with hemolysis reagent and blood glucose was measured in duplicate using the Sigma Enzymatic (Glucose Oxidase Trinder; ThermoFisher Microgenics, UK) colorimetric method and a SpectraMax 250 (Molecular Devices Corporation, Sunnyvale, CA, USA). Areas under the curve were analyzed from 0 to 120 min.

Insulin tolerance was measured after the mice had been fasted for five hours before being dosed with insulin (Actrapid™ Centaur Services, UK at 0.5 units/kg body weight, i.p.). Blood samples were taken, as described for the glucose tolerance test, 10 min and immediately before, and 10, 20, 30, 45 and 60 min after the administration of insulin.

Glucose uptake by adipocytes

At least 10 mice or two rats were used for each preparation of adipocytes. Tissue was minced and digested with collagenase type II in Krebs–Ringer HEPES buffer containing 10 mM HEPES, 1% bovine serum albumin (fraction V), 2.5 mM CaCl2, 5.5 mM glucose and 200 nM adenosine at pH 7.4 at 37 °C. The preparation was filtered through a 250–300 µm nylon mesh. The infranatant was removed and the floating layer of adipocytes was washed four times with fresh buffer. Adipocytes were concentrated to 40% of the final volume of Krebs–Ringer HEPES buffer containing 5% BSA and 0.3 mM glucose and pre-incubated for 45 min under 95% O2: 5% CO2 before dispensing them into 300 µl polyethylene tubes for the measurement of glucose uptake.

Glucose uptake was measured as described previously (Kashiwagi, Huecksteadt & Foley, 1983). Adipocytes were incubated in Krebs–Ringer HEPES buffer containing BSA, 0.3 mM glucose and D-[U-14C] glucose (0.2 µmol/l; 0.2 µCi/ml), for 1 h at 37 °C in the absence or presence of different concentrations of chemerin, or C-terminal chemerin peptides and insulin. The reaction was stopped by separation of the cells through silicone oil and radioactivity in the cells was measured. The extracellular space was measured in parallel incubations using D-[U-14C] sucrose. Uptake is expressed relative to the weight of the cells, or relative to uptake in the presence of insulin but absence of chemerin or its C-terminal peptides, as indicated in the figure legends. For each preparation of adipocytes there were at least four or five replicates of each treatment. Apart from the preliminary data shown in Fig. S8, the data presented are means of the replicate means from four or five preparations of adipocytes. Glucose uptake was calculated assuming that 2-deoxyglucose, and glucose are not distinguished by cellular uptake mechanisms. Further details are given in the figure legends.

Plasma chemerin assay

Plasma chemerin levels were measured using a murine chemerin quantikine ELISA kit (R and D Systems, Oxford, UK) according to the manufacturer’s recommendation. Whole blood was collected into EDTA tubes and spun at 3,000 g for 5 min at 4 °C and the plasma stored at −80 °C prior to analysis. Plasma samples were assayed in duplicate and the absorbance of both unknowns and standards measured using the Spectromax at 450 nm with a sensitivity range of 1.08–3.47 pg/ml.

Gene expression

Total RNA was isolated from white adipose tissue and analysed using a NanoDrop ND1000 (Thermo Fisher Scientific, Delaware, USA). Real time PCR (StepOne™, Applied Biosystems, Paisley, UK) was carried out using Assay-on-Demand pre-designed primer and probe sets for the chemerin gene, RARRES2, and GAPDH (Applied Biosystems, Paisley, UK). GAPDH was chosen as the housekeeping genes because it showed consistent CT values in adipose tissue. Data were analysed using the comparative ΔCt method. All procedures were carried out in accordance to the manufacturer’s recommendation.

Statistics

Unless stated differently in the text or legends (see Fig. 6), data were analyzed by one-way ANOVA followed by Fisher’s least significant difference test, using GraphPad Prism version 5 (GraphPad™ software; San Diego CA, USA). Two-sided significance levels are given. Differences were considered significant at P < 0.05. Results are expressed as means ± SEM.

Results

CMKLR1 knockout mice fed on a high-fat diet display sex- and age-dependent impaired glucose homeostasis and increased percentage body fat

With the exception of insulin tolerance, which was measured only at 6 months of age, and food intake and body weight, which were measured weekly, measurements were conducted at 7–9 weeks, 11–13 weeks and six months of age.

When they were fed on a standard chow diet, there were no differences in body weight between genotypes (Fig. S1), but the female knockout mice had a lower body fat content than the wild-type mice at the age of 6 months (Fig. S2). Both the knockout and heterozygote female mice had a lower fasting blood glucose than the wild-type mice before the glucose tolerance test at the age of 3 months (Fig. S3). There were no significant differences at any time between wild-type, heterozygote and knockout genotypes (either males or females) in food intake (results not shown), glucose tolerance, insulin levels during an oral glucose tolerance test, or blood glucose concentration following administration of insulin (Figs. S3, S4 and S5). There were no differences between genotypes in any of these measurements when the mice were 7–9 weeks old. The plasma chemerin concentration was higher in 3-month-old chow-fed male knockout mice (116 ± 17 ng/ml) than in wild-type (74 ± 4 ng/ml) or heterozygote (75 ± 7 ng/ml) mice (P < 0.05; n = 6; non-parametric Kruskal–Wallis test followed by Dunn’s multiple comparison test because the Bartlett test showed unequal variances).

A different picture emerged when the mice were fed on a high-fat diet. These were in the direction of increased adiposity or impaired glucose tolerance in the knockout or heterozygote mice compared to the wild-type mice. Thus, although there were no significant differences between genotypes in body weight (Fig. 1), both the male knockout and heterozygote mice had a higher percentage of body fat than the wild-type mice at 3 but not 6 months of age, whilst both the female knockout and heterozygote mice had a higher percentage of body fat at 6 but not 3 months of age (Fig. 2). These differences were mostly due to higher body fat content, except that the 3-month-old male heterozygote mice had less lean tissue than the wild-type mice, coupled with a statistically non-significant (P = 0.12) increase in body fat content. There were no differences in body composition between genotypes when the mice were 7–8 weeks old (results not shown).

Figure 1 Body weight of wild-type, CMKLR1 knockout and CMKLR1 heterozygote mice fed on a high fat diet.

n = 17–21 for males, 14–17 for wild-type and heterozygote females, and 8–10 for female knockout mice. The upper three lines are for males and the lower three for females.

Figure 2 Body composition of wild-type, CMKLR1 knockout and CMKLR1 heterozygote mice fed on a high-fat diet at 12 weeks (A, B, C) and six months (D, E, F) of age.

n = 17–21 for males, 14 and 17 for wild-type and heterozygote females at 12 weeks and 6 months respectively, and 8 and 10 for female knockout mice at 12 weeks and 6 months respectively. ∗P < 0.05 for knockout or heterozygote compared to wild-type mice of the same sex.

The male knockout mice fed on a high fat diet had a higher fasting blood glucose than the wild-type controls at 9 weeks and at 3 months of age (Fig. 3), and a higher blood glucose level 10 min after administration of insulin at 6 months of age (Fig. 4), despite body composition being no different from that of wild-type mice at 6 months of age. The male heterozygote mice had elevated plasma glucose at 3 months of age and an elevated fed blood glucose at both 3 and 6 months of age (Fig. 3). Oral glucose tolerance and plasma insulin during the glucose tolerance test was unaffected by genotype at all ages in both the male knockout and male heterozygote mice (Fig. 4; Figs. S6 and S7).

Figure 3 Blood glucose concentration in 5 h-fasted (A, C, E) and fed (B, D, F) wild-type, CMKLR1 knockout and CMKLR1 heterozygote mice fed on a high-fat diet.

(A, B) 8–9 weeks, (C, D) 13 weeks and (E, F) six months of age. n = 17–21 at 8–9 weeks, and 12–15 at 13 weeks and six months of age, except for female knockout mice, where n = 5–10. ∗P < 0.05; ∗∗P < 0.01 for knockout or heterozygote compared to wild-type mice of the same sex.

Figure 4 Oral glucose tolerance and intraperitoneal insulin tolerance in wild-type, CMKLR1 knockout and CMKLR1 heterozygote mice fed on a high-fat diet.

Glucose tolerance curves in 12-week-old male (A) and female (B) mice. Areas under the glucose tolerance curves in 12-week-old (C) and 6-month-old (D) mice. Insulin tolerance curves in 6-month-old male (E) and female (F) mice. Further data from the oral glucose tolerance tests are given in Figs. S3–S7. n = 17–21, except for female knockout mice, where n = 10. ∗P < 0.05 for knockout or heterozygote compared to wild-type mice of the same sex.

The female knockout mice fed on a high-fat diet had a higher fasting plasma glucose concentration than the wild-type mice (Fig. 3) and a higher blood glucose level 10 min after administration of insulin (Fig. 4). At this age (6 months), they had higher body fat content. Their glucose tolerance was impaired at 3 months of age, despite their body composition being no different from that of wild-type mice (Fig. 2). The female heterozygote mice had a higher blood glucose level than the wild-type mice 10 min after administration of insulin at 6 months of age (Fig. 4).

Chemerin and its C-15 and C-19 terminal peptides stimulate glucose uptake by adipocytes

Initial experiments (Fig. S8) using murine mesenteric adipocytes suggested that chemerin and its C-15 and C-19 terminal peptides (Cash et al., 2008), but not the C-9 analogue peptide (Shimamura et al., 2009), stimulated glucose uptake in the presence of 0.1 nM insulin, which had no effect alone, but not in the absence of insulin or in the presence of 10 nM insulin. Higher concentrations than 1 nM chemerin appeared less effective, a phenomenon that has been observed for other effects of chemerin (Bozaoglu et al., 2010; Cash et al., 2008) and the C-15 peptide (Cash, Christian & Greaves, 2010).

We were able to generate more robust data by using rat epididymal adipocytes rather than murine epididymal adipocytes. Studies were conducted on the C-9 analogue, C-15 and C-19 fragments using four separate adipocyte preparations to ensure that the results were not a peculiarity of one preparation. The C-9 analogue was ineffective, C-15 had a similar potency to chemerin with a peak effect at 40 pM, whilst C-19 had a similar peak effect but at the higher concentration of 160 pM (Fig. 5A).

Four separate rat epididymal adipocyte preparations were then used to compare the effects of 40 pM chemerin and C-15 on the insulin concentration-response curve. The C-15 peptide shifted the insulin curve further to the left than did full length recombinant chemerin (Fig. 5B). Chemerin and C-15 were ineffective in the presence of high concentrations of insulin.

Figure 5 Effect of murine chemerin and C-terminal peptides on glucose uptake by rat epididymal adipocytes.

(A) Glucose uptake in the presence of 0.05 nM insulin in response to 10, 100 or 1,000 pM of chemerin, C15-terminal peptide, C19-terminal peptide or the stable analogue of the C9-terminal peptide. ∗∗P < 0.01; ∗∗∗P < 0.001 for knockout or heterozygote compared to wild-type mice. (B) Glucose uptake in the absence or presence of murine chemerin (40 pM) or its C15-terminal peptide in response to insulin. Results in each panel are means of four experiments on different preparations of adipocytes. In each experiment, each data point was the mean of five replicates. In (A), results for each experiment were expressed relative to the mean of the five replicates for uptake in the presence of insulin alone.

The stimulatory effect of the C-15 fragment of chemerin is partly mediated by CMKLR1

Having established that the C-15 peptide had the greatest effect on glucose uptake in rat epididymal adipocytes, the effect of this peptide (40 pM) was compared in murine epididymal adipocytes from wild-type and knockout mice. The peptide increased insulin-stimulated glucose uptake in adipocytes from wild-type, but not in knockout mice. A similar result was obtained with full length recombinant murine chemerin, but the effect of chemerin in the wild-type adipocytes did not achieve statistical significance in this experiment (Fig. 6).

Figure 6 Glucose uptake in the presence of insulin (0.3 nM) in response to murine chemerin (40 pM) or its C15-terminal peptide (40 pM) in epididymal adipocytes from wild type and CMKLR1 knockout mice.

Results in each panel are means of five experiments on different preparations of adipocytes. In each experiment, each data point was the mean of five replicates expressed relative to the mean of the five replicates for uptake in the presence of insulin alone. The effects of full length chemerin and its C15-terminal peptide in wild-type versus knockout adipocytes were compared by unpaired t-tests using Welch’s correction to allow for unequal variances. ∗∗P < 0.01 for the effect of C15-terminal peptide on glucose uptake in wild-type versus CMKLR1 knockout adipocytes. The effects of chemerin and its C15-terminal peptide in the wild-type adipocytes were analysed by repeated measures (repeated preparations of adipocytes) 2-way ANOVA. †P < 0.01 for the effect of the C15-terminal peptide.

The plasma chemerin concentration is raised in obesity and decreased by fasting

FVB mice are known to be less susceptible than C57Bl/6 mice to diet-induced obesity (Glendinning et al., 2010; Kim et al., 2013), as we confirmed (Fig. 7A; Fig. S9). Plasma chemerin levels were raised by feeding male FVB or C57Bl/6 mice on a high fat diet for 6 months (Fig. 7B). The increase in plasma chemerin levels in response to the high-fat diet in the FVB mice was 75% of that in the C57Bl/6 mice, whereas the increases in body fat content and percentage body fat in FVB mice were 36% and 57% respectively of those in C57Bl/6 mice. The expression of RARRES2, the chemerin gene, was higher in inguinal adipose tissue in high fat-compared to chow-fed C57Bl/6 mice (Fig. 7D). There was no difference in expression in epididymal adipose tissue (results not shown).

Figure 7 Effects of obesity and fasting on the plasma chemerin concentration and chemerin gene (RRARES2) expression in inguinal adiposetissue.

The effect of feeding on a high-fat diet for 6 months from weaning on body fat mass (A; n = 7 to 9) and plasma chemerin concentration (B; n = 5 to 8) in male C57Bl6 and FVB mice. Plasma chemerin (C; n = 5 to 6) and RARRES2 mRNA concentration (D; n = 5) in inguinal tissue in fed and 5 h-fasted C57BL/6 8-week-old wild-type and 10-week-old ob/ob mice (as in C), and in chow- and high fat-fed C57Bl/6 mice (as in B). ∗P < 0.05; ∗∗P < 0.01; ∗∗∗P < 0.001 for the comparisons indicated by the bars.

The concentration of chemerin in the plasma of fed male ob/ob mice was higher than in wild-type controls (Fig. 7C). We cannot exclude the possibility that this is because the ob/ob mice were two weeks older than the wild-type mice. However, by contrast, the expression of RARRES2 was lower in inguinal adipose tissue in ob/ob than in C57Bl/6 mice (Fig. 7D). Other workers have also noted this paradox (Ernst et al., 2010). We found no difference in the plasma chemerin level or the expression of RARRES2 in inguinal adipose tissue between male db/db mice and wild-type mice of the background (C57Bl/Ks) strain (Fig. S10).

Plasma chemerin was reduced by fasting in both wild-type male C57Bl/6 and female ob/ob mice (Figs. 7C and 8A). The expression of RARRES2, the chemerin gene, was similarly lower in inguinal adipose tissue of fasted than fed wild-type mice (Fig. 7D).

Fasting or feeding on a high fat diet did not alter the expression of RARRES2 in epididymal adipose tissue (Fig. S11).

Rosiglitazone increases the plasma chemerin concentration in ob/ob mice

The plasma chemerin concentration was increased by treatment of female ob/ob mice with rosiglitazone (3 mg/kg body weight, p.o.) for 18 days (following an overnight fast) or 21 days (in the fed state; Fig. 8A). Treatment of female ob/ob mice with rosiglitazone increased the expression of RARRES2 in inguinal (Fig. 8B) but not parametrial white adipose tissue (Fig. S11B).

Figure 8 The effect of fasting and rosiglitazone on plasma chemerin concentration (A) and RARRES2 expression in inguinal adipose tissue (B) in female C57Bl/6J ob/ob mice.

Mice were dosed with rosiglitazone (3 mg/kg body weight, p.o. daily) for three weeks. Plasma chemerin was measured before the 19th dose following a 16 h overnight fast (n = 7) and at termination in the fed state (n = 8). RARRES2 expression was measured in the fed state (n = 5). ∗P < 0.05; ∗∗P < 0.01; ∗∗∗P < 0.001 for the comparisons indicated by the bars.

Discussion

The finding that plasma chemerin levels are raised in human obesity (Bozaoglu et al., 2007; Bozaoglu et al., 2009) and in genetically obese ob/ob mice (Ernst et al., 2010), the adipogenic effect of chemerin (Goralski et al., 2007), and a report of decreased percentage body fat in mice that lack the chemerin receptor CMKLR1 (Ernst et al., 2012) have led to suggestions that raised plasma chemerin levels promote obesity. Other findings have led to suggestions that chemerin is in part responsible for the link between obesity and insulin resistance. Thus, raised plasma chemerin is associated with the metabolic syndrome (Stejskal et al., 2008; Bozaoglu et al., 2007) and chemerin is released from adipocytes (Goralski et al., 2007). The majority of studies find chemerin to be pro-inflammatory (Ernst & Sinal, 2010). Chemerin inhibited insulin action in skeletal muscle cells (Sell et al., 2009) and in one study in 3T3 L1 adipocytes (Kralisch et al., 2009). Administration of chemerin exacerbated glucose tolerance in mice that are obese and diabetic (Ernst et al., 2010), and mice that lack the chemerin receptor are susceptible to diet-induced insulin resistance (Ernst et al., 2012). These results support the view that an antagonist of chemerin action, in particular of CMKLR1, might be useful in the treatment of type 2 diabetes associated with obesity.

The findings that we report here lead, however, to a different conclusion—that chemerin, acting through CMKLR1, opposes diet-induced obesity and insulin resistance, and that therefore an agonist of CMKLR1 might be of value in the treatment of type 2 diabetes.

First, although our six-month-old female CMKLR1 knockout mice had a lower body fat content than wild-type mice when they were fed on a low-fat diet, knockout and heterozygote mice of both sexes had a slightly higher percentage body fat than wild-type mice at three or six months of age when they were fed on a high-fat diet.

Second, when fed on the high fat diet, fasting glucose was higher in our knockout than in our wild-type mice when males were nine weeks or three months old, and females were six months old. Fasting blood glucose was also raised in three-month-old heterozygote male mice. Glucose tolerance was impaired in the female knockout mice when they were three months old, and there was some impairment in insulin tolerance in both male and female six-month-old knockout mice.

Overall, our results suggest that CMKLR1 knockout and heterozygote mice tend to be mildly obese and have impaired glucose homeostasis when they are fed on a high-fat diet. The impaired glucose homeostasis does not appear to be entirely a consequence of the obesity, because it sometimes occurred in the absence of obesity. Chemerin stimulates adipogenesis and angiogenesis (Bozaoglu et al., 2010; Goralski et al., 2007), so the possibility that its absence may restrict the ability of adipocytes to expand and act as a ‘sink’ for glucose was considered as a mechanism for impaired glucose homeostasis in the absence of obesity. However, this hypothesis is inconsistent with the tendency of high fat-fed knockout and heterozygote mice to have a higher body fat content than wild-type mice.

Previous studies have shown an inconsistent metabolic phenotype for CMKLR1 mice. The first study reported was conducted in male mice (sex of the mice from personal communication from Professor Christopher Sinal). These mice displayed a lean phenotype when fed on either a low- or high-fat diet. In mice fed on a high-fat diet, leanness was associated with lower fasting plasma glucose and serum insulin. Surprisingly in view of their leanness, the mice displayed impaired glucose tolerance, regardless of diet. This was associated with decreased insulin secretion, as there was no alteration of insulin tolerance (Ernst et al., 2012). In another study, rather than being lean, male but not female mice that lacked CMKLR1 developed increased adipose tissue mass from the age of 8 months when fed on a standard (3.5% fat) diet, but glucose tolerance was unchanged. However, when the mice were fed on a high-fat diet for 20 weeks, there was no difference in body weight between the knockout and wild-type male mice (Rouger et al., 2013). In a third study, in which male mice were fed on a high-fat diet, the absence of CMKLR1 had no effect on body weight, food intake or insulin sensitivity (Gruben et al., 2014).

Our results agree with reports that plasma chemerin levels are raised in obese rodents. Thus, they were higher in ob/ob than in wild-type mice as reported by others (Ernst et al., 2010). We also found that they are raised in diet-induced obese mice, which is more representative of human obesity, and that they were reduced by overnight fasting. It might be argued that, since chemerin has been found to stimulate adipogenesis (Goralski et al., 2007; Roh et al., 2007), raised plasma levels would exacerbate obesity. This view is not, however, supported by comparing our results in FVB and C57Bl/6 mice. Plasma chemerin levels were raised by feeding either FVB or C57Bl/6 mice on a high-fat diet. The increase in the chemerin level in the FVB mice was about 75% of that in the C57Bl/6 mice, whereas the increase in the body fat content in FVB mice was about 30% of that in C57Bl/6 mice. This might be interpreted in terms of increased plasma chemerin protecting the FVB mice from diet-induced obesity, rather than it promoting obesity through its adipogenic activity. By contrast with our results, others have reported that the plasma chemerin concentration was not raised by feeding NMRI mice on a high-fat or cafeteria diet for three weeks, despite these treatments roughly doubling weight gain (Hansen et al., 2014). NMRI mice are less prone than C57Bl/6 mice to diet-induced obesity (Matyskova et al., 2008).

If chemerin does provide some protection against obesity, it does not seem that this is directly linked to increased expression of RARRES2 mRNA in perigenital adipose tissue. Thus, we did not find any changes in the expression of RARRES2 in response to high fat feeding or fasting in perigenital adipose tissue. By contrast, plasma chemerin levels paralleled the expression of RARRES2 in inguinal adipose tissue, except that expression of RARRES2 was lower in inguinal adipose tissue of ob/ob mice than in wild-type mice, despite plasma chemerin concentrations being higher. Other workers have reported increased expression of RARRES2 in epididymal adipose tissue in mice fed on a high-fat diet (Roh et al., 2007), and decreased expression in rat perirenal, inguinal and epididymal adipose tissue (though only perirenal results are shown) in response to fasting for 72 h or food restriction by 50% for one month (Stelmanska et al., 2013). However, our results resemble those of Hansen et al., who found that that cold acclimation reduced and obesogenic diets increased RARRES2 expression in inguinal but not epididymal adipose tissue. Expression in interscapular brown adipose tissue paralleled that in inguinal adipose tissue, the latter site (like the perirenal site) being one where brite/beige adipocytes are expressed (Hansen et al., 2014).

The plasma chemerin concentration has been reported to be higher in db/db mice than in wild-type C57Bl/6 mice, which is the background strain for ob/ob mice (Ernst et al., 2010). We did not find that the plasma chemerin concentration was higher in db/db mice relative to wild-type mice of its own background strain (C57Bl/Ks), though there was a slight trend in this direction.

Treatment of ob/ob mice with the insulin sensitising drug rosiglitazone increased their plasma chemerin concentration and the expression of RARRES2 in inguinal but not parametrial adipose tissue. Others have also reported that rosiglitazone increased RARRES2 mRNA in inguinal adipose tissue of normal mice and identified a PPARγ response element within the RARRES2 promoter (Muruganandan et al., 2011). It is well-established that rosiglitazone and other PPARγ agonists have anti-inflammatory properties (Ceriello, 2008). Therefore, the effects of rosiglitazone in our study are more consistent with chemerin having anti-inflammatory and anti-diabetic than pro-inflammatory and pro-diabetic activity. However, we note that other PPARγ agonists have been reported to reduce RARRES2 mRNA levels in epididymal and mesenteric adipose tissue when administered to mice (Vernochet et al., 2009), and plasma chemerin levels when administered to patients with type 2 diabetes (Esteghamati et al., 2014).

It is possible that the effect of rosiglitazone on RARRES2 mRNA levels is unrelated to its anti-inflammatory activity but due to its promotion of brown adipocyte formation, whilst possibly decreasing sympathetic activity (Sell et al., 2004). This seems unlikely, however, because cold acclimation and obesogenic diets, both of which increased brown adipose tissue formation, did not affect the plasma chemerin concentration. Moreover, cold acclimation reduced, whereas obesogenic diets increased, RARRES2 mRNA levels in interscapular brown and inguinal brite/beige adipose tissue (Hansen et al., 2014).

We investigated whether chemerin might have a direct insulin sensitising effect in primary adipocytes. This had been suggested by a previous study using 6 nM chemerin in 3T3-L1 adipocytes (Takahashi et al., 2008), although a study using an extremely high concentration (10 µM) of chemerin found reduced insulin-stimulated uptake (Kralisch et al., 2009). In addition to chemerin, we studied some C-terminal fragments of chemerin because we were interested in whether they might provide the basis for a drug discovery programme. The C-terminal nonapeptide has been identified as being the smallest peptide that had functional activity at low nanomolar concentration in a Chinese hamster ovary cell line that expressed RARRES2 (Wittamer et al., 2004) and a stable analogue with similar potency has been described (Shimamura et al., 2009). This analogue did not, however, stimulate glucose uptake by mouse mesenteric or rat epididymal adipocytes, either in the absence or presence of insulin. By contrast, the C-15 and C-19 terminal peptides (Cash et al., 2008) and the full length chemerin itself appeared to stimulate glucose uptake by mouse mesenteric adipocytes in the presence, but not the absence, of a submaximal concentration of insulin, and this property was more clearly demonstrated using rat epididymal adipocytes. The peak effects of chemerin and the C-15 peptide were at 40 pM, and at this concentration both full length chemerin and the C-15 peptide shifted the chemerin concentration-response curve to the left. Using mouse epididymal adipocytes from knockout and wild-type mice, we were able to show that CMKLR1 mediated the response to the C-15 peptide and probably to chemerin as well, although unfortunately the effect of chemerin did not achieve statistical significance in that experiment. CMKLR1 has also been shown to mediate the anti-inflammatory effect of the C-15 peptide (Cash et al., 2008).

We attempted to investigate the potential of agonists of CMKLR1 in the treatment of obesity using chemerin and the C-terminal peptides described here, by following the approach used by others to demonstrate that the stable C-9 analogue is active in vivo. In that work the plasma non-esterified fatty acid concentration was reduced 30 and 60 min following intraperitoneal injection of a single dose of this peptide (30 mg/kg body weight) into overnight fasted mice, though it was not demonstrated that the effect was mediated by CMKLR1 (Shimamura et al., 2009). We, however, found no effect of chemerin or any of the peptides on non-esterified fatty acid or glucose concentrations in mice fed on low- or high-fat diets (results not shown). The highest doses employed in our studies were 10 µg (about 0.25 mg/kg body weight) of murine chemerin and the stable C-9 analogue in mice that had been made obese by feeding them on a high-fat diet. This dose of chemerin would have been sufficient to at least double the plasma chemerin concentration if it were stable and evenly distributed, but no increase was detected 30 or 60 min after its injection, consistent with its being rapidly metabolised. The failure to demonstrate an effect of the C-9 analogue is consistent with its lack of effect on CMKLR1-mediated glucose uptake, but it is possible that a higher dose would have been effective. Demonstration of proof-of-concept in an animal model of type 2 diabetes and obesity clearly requires the identification of a stable compound that is more effective as an agonist of CMKLR1.

In conclusion, knockout and heterozygote mice for the chemerin CMKLR1 receptor tend to be mildly obese and display mild impairment of glucose homeostasis. Rosiglitazone increases the plasma chemerin concentration in ob/ob mice, suggesting that chemerin might be anti-inflammatory and anti-diabetic. Chemerin and its C-15 terminal peptide, acting via the CMKLR1 receptor, sensitise adipocytes to the stimulatory effect of insulin on glucose uptake. These findings suggest that an agonist of CMKLR1 might be useful for the treatment of type 2 diabetes associated with obesity.

Supplemental Information

Figure S1 Body weight of wild-type, CMKLR1 knockout and CMKLR1 heterozygote mice fed on a standard chow diet

n = 15 for wild-type and heterozygote mice, and 11 and 7 for male and female knockout mice respectively. The upper three lines are for males and the lower three for females.

Click here for additional data file.

Figure S2 Body fat content of wild-type, CMKLR1 knockout and CMKLR1 heterozygotemice fed on a standard chow diet at 12 weeks (A) and six months (B) of age

n values are as for supplementary Fig. 1.

Click here for additional data file.

Figure S3 Oral glucose tolerance and intraperitoneal insulin tolerance in wild-type, CMKLR1 knockout and CMKLR1 heterozygote mice fed on a standard chow diet at 12 weeks (A) and six months (B) of age

n values are as for Supplementary Fig. 1. ∗P < 0.05 for knockout and heterozygote compared to wild-type mice of the same sex.

Click here for additional data file.

Figure S4 Plasma insulin during the oral glucose tolerance test in wild-type, CMKLR1 knockout and CMKLR1 heterozygote mice fed on a standard chow diet at 12 weeks (A) and six months (B) of age

n values are as for Supplementary figure 1.

Click here for additional data file.

Figure S5 Insulin tolerance test in 6-month-old wild-type, CMKLR1 knockout and CMKLR1 heterozygote mice fed on a standard chowdiet

n values are as for Supplementary figure 1.

Click here for additional data file.

Figure S6 Oral glucose tolerance and intraperitoneal insulin tolerance in 6-month-old wild-type, CMKLR1 knockout and CMKLR1 heterozygote mice fed on a high fat diet

n = 17 to 21, except for female knockout mice, where n = 10.

Click here for additional data file.

Figure S7 Plasma insulin during the oral glucose tolerance test in wild-type, CMKLR1 knockout and CMKLR1 heterozygote mice fed on a high fat diet at 12 weeks (A) and six months (B) ofage

Click here for additional data file.

Figure S8 Preliminary experiments on the effects of murine chemerin, its C-15 and C-19 terminal peptides and a stable analogue of its C9-terminal peptide on glucose uptake inmurine mesenteric adipocytes

Panel (A) shows consolidated results for two preparations of adipocytes which gave very similar basal values and where concentrations of chemerin were identical between experiments; n = 4 to 10. Panels(B) and (C) show means for 4 replicates from single preparations of adipocytes.

Click here for additional data file.

Figure S9 Body weights (A) and percentage bodyfat (B) of male FVB and C57Bl/6 mice fed on chow and high fat diets for 6 months

n = 7 to 9. ∗P < 0.05; ∗∗P < 0.01; ∗∗∗P < 0.001 for the comparisons indicated by the bars.

Click here for additional data file.

Figure S10 Plasma chemerin concentration (A) and RARRES2 mRNA levels (B) in inguinal adipose tissue of 11- to 12-week-old male db/db mice and the background C57Bl/Ks strain

n = 6 in A and 5 in B.

Click here for additional data file.

Figure S11 RARRES2 mRNA in perigenital adipose tissue

(A) Fed and 5 h-fasted C57BL/6 8-week-old wild-type and 10-week-old ob/ob mice, and (B) female ob/ob mice treated with rosiglitazone for 3 weeks. n values and other details are given in the legends to Fig. 7D and 8B of the main paper, which, by contrast, show changes in RARRES2 expression in inguinal fat.

Click here for additional data file.

The authors are grateful to Mrs Anita Roberts and Mr David Hislop for their technical support.

Additional Information and Declarations

Competing Interests

Author Contributions

Animal Ethics

Steven J. Wang was an employee at AstraZeneca at the time of these studies. AstraZeneca is a pharmaceutical company engaged in the marketing, development and discovery of drugs to treat diabetes.

Edward T. Wargent performed the experiments, analyzed the data, contributed reagents/materials/analysis tools, prepared figures and/or tables.

Mohamed S. Zaibi performed the experiments, analyzed the data, contributed reagents/materials/analysis tools, prepared figures and/or tables, reviewed drafts of the paper.

Jacqueline F. O’Dowd performed the experiments, analyzed the data, contributed reagents/materials/analysis tools.

Michael A. Cawthorne reviewed drafts of the paper.

Steven J. Wang conceived and designed the experiments, contributed reagents/materials/analysis tools, reviewed drafts of the paper.

Jonathan R.S. Arch conceived and designed the experiments, analyzed the data, wrote the paper, prepared figures and/or tables, reviewed drafts of the paper.

Claire J. Stocker conceived and designed the experiments, performed the experiments, analyzed the data, contributed reagents/materials/analysis tools, wrote the paper, prepared figures and/or tables, reviewed drafts of the paper.

The following information was supplied relating to ethical approvals (i.e., approving body and any reference numbers):

1. The University of Buckingham Ethical review Board. 2. Bu08-039; Bu09-007; Bu 11-001; Bu11-009.

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
