# Peer review of "Evidence from studies in rodents and in isolated adipocytes that agonists of the chemerin receptor CMKLR1 may be beneficial in the treatment of type 2 diabetes"

_PeerJ, doi:10.7717/peerj.753_

## Round 0.1 · original submission · Major Revisions

Both reviewers feel this work is of interest, however, more explanation on the "chemerin pathway" and a concise presentation are desired.

Reviewer 1 ·

Basic reporting

No Comments

Experimental design

It is important to perform the following experiments.
1. Compare the plasma chemerin level in wild-type, CMKLR1 knockout, and heterozygote mice.
2. Determine whether murine chemerin and some C-terminal peptides would decrease the high fat diet-induced obesity as well as glucose intolerance, through CMKLR1 or not.

Validity of the findings

Without showing the data mentioned above, the conclusion of this study is remarkably weakened. Please consider to supplement the manuscript with these data.

Additional comments

This study addresses an interesting and important question. Knockout mouse models used in this study are valuable. The authors are encouraged to provide more explanation and insights on the regulation of the "chemerin pathway" therapeutically. Specifically, is it on the level of agonists or the expression of chemerin receptor?

Reviewer 2 ·

Basic reporting

No Comments

Experimental design

Experimental designed include too many conditions with superficial observation, which made this study without a clear focus and deep mechanistic insight.

Validity of the findings

No Comments

Additional comments

There were hug data, but the associations of chemerin with studied conditions were not very excited since almost of measurements were mildly changed.

---

## Round 0.2 · accepted · Accept

The authors have carefully addressed reviewers question and revised the manuscript according by adding new data and discussion. This work is important to show that a CMKLR1 agonist might improve glucose homeostasis by increasing insulin-stimulated glucose uptake by adipocytes and would be of benefit in the treatment of type 2 diabetes associated with obesity.